# Identification of novel and rare variants associated with handgrip strength using whole genome sequence data from the NHLBI Trans-Omics in Precision Medicine (TOPMed) Program

**Chloé Sarnowski**[1,2]*, **Han Chen**[2,3], **Mary L. Biggs**[4,5], **Sylvia Wassertheil-Smoller**[6], **Jan Bressler**[2], **Marguerite R. Irvin**[7], **Kathleen A. Ryan**[8], **David Karasik**[9,10], **Donna K. Arnett**[11], **L. Adrienne Cupples**[1,12], **David W. Fardo**[13], **Stephanie M. Gogarten**[5], **Benjamin D. Heavner**[5], **Deepti Jain**[5], **Hyun Min Kang**[14], **Charles Kooperberg**[15], **Arch G. Mainous**[16], **Braxton D. Mitchell**[8,17], **Alanna C. Morrison**[2], **Jeffrey R. O'Connell**[8], **Bruce M. Psaty**[4,18,19], **Kenneth Rice**[5], **Albert V. Smith**[14], **Ramachandran S. Vasan**[12,20,21], **B. Gwen Windham**[22], **Douglas P. Kiel**[9,23,24], **Joanne M. Murabito**[12,25], **Kathryn L. Lunetta**[1], on behalf of the TOPMed Longevity and Healthy Aging Working Group[¶], from the NHLBI Trans-Omics for Precision Medicine (TOPMed) Consortium[¶]

1 Department of Biostatistics, Boston University School of Public Health, Boston, MA, United States of America, 2 Human Genetics Center, Department of Epidemiology, Human Genetics and Environmental Sciences, School of Public Health, The University of Texas Health Science Center at Houston, Houston, TX, United States of America, 3 Center for Precision Health, School of Public Health and School of Biomedical Informatics, The University of Texas Health Science Center at Houston, Houston, TX, United States of America, 4 Cardiovascular Health Unit, Department of Medicine, University of Washington, Seattle, WA, United States of America, 5 Department of Biostatistics, University of Washington, Seattle, WA, United States of America, 6 Department of Epidemiology and Population Health, Albert Einstein College of Medicine, Bronx, NY, United States of America, 7 Department of Epidemiology, University of Alabama at Birmingham School of Public Health, Birmingham, AL, United States of America, 8 Division of Endocrinology, Diabetes and Nutrition, Department of Medicine, University of Maryland School of Medicine, Baltimore, MD, United States of America, 9 Hinda and Arthur Marcus Institute for Aging Research, Hebrew SeniorLife, Boston, MA, United States of America, 10 Azrieli Faculty of Medicine, Bar Ilan University, Safed, Israel, 11 University of Kentucky, College of Public Health, Lexington, KY, United States of America, 12 National Heart Lung and Blood Institute and Boston University's Framingham Heart Study, Framingham, MA, United States of America, 13 Department of Biostatistics and Sanders-Brown Center on Aging, University of Kentucky, Lexington, KY, United States of America, 14 Department of Biostatistics, University of Michigan, Ann Arbor, MI, United States of America, 15 Division of Public Health Sciences, Fred Hutchinson Cancer Research Center, Seattle, WA, United States of America, 16 Department of Health Services Research, Management and Policy, University of Florida, Gainesville, FL, United States of America, 17 Geriatrics Research and Education Clinical Center, Baltimore Veterans Administration Medical Center, Baltimore, MD, United States of America, 18 Departments of Epidemiology and Health Services, University of Washington, Seattle, WA, United States of America, 19 Kaiser Permanente Washington Health Research Institute, Seattle, WA, United States of America, 20 Section of Preventive Medicine and Epidemiology, Evans Department of Medicine, Boston University School of Medicine, Boston, MA, United States of America, 21 Whitaker Cardiovascular Institute and Cardiology Section, Evans Department of Medicine, Boston University School of Medicine, Boston, MA, United States of America, 22 The MIND Center, University of Mississippi Medical Center, Jackson, MS, United States of America, 23 Department of Medicine, Beth Israel Deaconess Medical Center, Harvard Medical School, Boston, MA, United States of America, 24 Broad Institute of Harvard & MIT, Cambridge, MA, United States of America, 25 Section of General Internal Medicine, Department of Medicine, Boston University School of Medicine, Boston, MA, United States of America

¶ Membership of the TOPMed Longevity and Healthy Aging Working Group is provided in the Acknowledgments. NHLBI TOPMed Banner Authorship is provided as a S3 File.
* Chloe.Sarnowski@uth.tmc.edu

**Data Availability Statement:** The data used in this study are available through dbGaP but with restrictions as approved by the individual study IRBs as these data contain potentially identifying and sensitive patient information. Requesting individual(s) must have IRB approval to use the data, and obtain a letter of collaboration from each study's principal investigator. The NHLBI Data Access Committee (nhlbigeneticdata@nhlbi.nih. gov) handles data requests for the TOPMed study accessions via the dbGaP Data Access Request system. The summary statistics from the association analyses that support the findings of this study will be posted on the TOPMed Genomic Summary Results dbGaP accession phs001974, "NHLBI TOPMed: Genomic Summary Results for the Trans-Omics for Precision Medicine Program" and made accessible, via controlled access and a General Research Use (GRU) consent designation, to applicants via the normative dbGaP application process.

**Funding:** TOPMed Whole genome sequencing (WGS) for the Trans-Omics in Precision Medicine (TOPMed) program was supported by the National Heart, Lung and Blood Institute (NHLBI). WGS for NHLBI TOPMed: FHS (phs000974) was performed at the Broad Institute of MIT and Harvard (3U54HG003067-12S2). WGS for NHLBI TOPMed: CHS (phs001368) was performed at the Baylor College of Medicine Human Genome Sequencing Center (HHSN268201600033I). WGS for NHLBI TOPMed: Amish (phs000956) was performed at the Broad Institute of MIT and Harvard (3R01HL121007-01S1). WGS for NHLBI TOPMed: ARIC (phs001211) was performed at the Baylor College of Medicine Human Genome Sequencing Center (3U54HG003273-12S2, HHSN268201500015C) and the Broad Institute for MIT and Harvard (3R01HL092577-06S1). WGS for NHLBI TOPMed: WHI (phs001237) was performed at the Broad Institute of MIT and Harvard (HHSN268201500014C). WGS for NHLBI TOPMed: HyperGEN (phs001293) was performed at the University of Washington Northwest Genomics Center (3R01HL055673-18S1). The Genome Sequencing Program (GSP) was funded by the National Human Genome Research Institute (NHGRI), the National Heart, Lung, and Blood Institute (NHLBI), and the National Eye Institute (NEI). The GSP Coordinating Center (U24 HG008956) contributed to cross-program scientific initiatives and provided logistical and general study coordination. ARIC is part of the Centers for Common Disease Genomics (CCDG) program, a large-scale genome sequencing effort to identify rare risk and protective alleles that contribute to a range of common disease phenotypes. The CCDG

# Abstract

Handgrip strength is a widely used measure of muscle strength and a predictor of a range of morbidities including cardiovascular diseases and all-cause mortality. Previous genome-wide association studies of handgrip strength have focused on common variants primarily in persons of European descent. We aimed to identify rare and ancestry-specific genetic variants associated with handgrip strength by conducting whole-genome sequence association analyses using 13,552 participants from six studies representing diverse population groups from the Trans-Omics in Precision Medicine (TOPMed) Program. By leveraging multiple handgrip strength measures performed in study participants over time, we increased our effective sample size by 7–12%. Single-variant analyses identified ten handgrip strength loci among African-Americans: four rare variants, five low-frequency variants, and one common variant. One significant and four suggestive genes were identified associated with handgrip strength when aggregating rare and functional variants; all associations were ancestry-specific. We additionally leveraged the different ancestries available in the UK Biobank to further explore the ancestry-specific association signals from the single-variant association analyses. In conclusion, our study identified 11 new loci associated with handgrip strength with rare and/or ancestry-specific genetic variations, highlighting the added value of whole-genome sequencing in diverse samples. Several of the associations identified using single-variant or aggregate analyses lie in genes with a function relevant to the brain or muscle or were reported to be associated with muscle or age-related traits. Further studies in samples with sequence data and diverse ancestries are needed to confirm these findings.

# Introduction

Handgrip strength is an accessible measure of muscle strength and is used as a proxy of muscular fitness. It is quantified by measuring the amount of static force that the hand can squeeze around a dynamometer. Handgrip strength varies by age, and men have greater handgrip strength than women at all ages [1, 2]. For both men and women, peak handgrip strength is observed in the fourth decade followed by a gradual decline with age. Handgrip strength is a predictor of a range of morbidities including cardiovascular disease and all-cause mortality [3], is a marker of frailty [4, 5] and a key component of sarcopenia [6]. Handgrip strength is considered a marker of healthy aging [7–9]. It has been reported to be associated with both overall and exceptional survival in men [10, 11] and may also be a marker of brain health [12–14].

Heritability of handgrip strength, estimated by twin studies, is high ($h^2$ = 30–65%) [15–17]. However, few genome-wide association studies (GWAS) have been conducted for handgrip strength. A large and well-powered handgrip strength GWAS (N = 195,180 European participants) combined data from the Cohorts for Heart and Aging Research in Genomic Epidemiology (CHARGE) Consortium and the UK Biobank (UKBB) and detected 16 genome-wide significant ($P<5\times10^{-8}$) loci associated with handgrip strength [18]. A more recent and larger GWAS was performed in 223,315 participants from the UKBB but analyzed a different phenotype, the relative handgrip strength, defined as the average of measurements of right and left hand divided by weight [19]. The handgrip strength GWAS conducted to date included

program was supported by NHGRI and NHLBI, and whole genome sequencing was performed at the Baylor College of Medicine Human Genome Sequencing Center (UM1 HG008898 and R01HL059367). Centralized read mapping and genotype calling, along with variant quality metrics and filtering were provided by the TOPMed Informatics Research Center (3R01HL-117626-02S1; contract HHSN268201800002I). Phenotype harmonization, data management, sample-identity QC, and general study coordination were provided by the TOPMed Data Coordinating Center (3R01HL-120393-02S1; contract HHSN268201800001I). We gratefully acknowledge the studies and participants who provided biological samples and data for TOPMed. The Analysis Commons was funded by R01HL131136. Study-specific acknowledgements FHS: The FHS acknowledges the support of contracts N01-HC-25195 and HHSN268201500001I from the National Heart, Lung and Blood Institute and grant supplement R01 HL092577-06S1 for this research. We also acknowledge the dedication of the FHS study participants without whom this research would not be possible. CHS: This research was supported by contracts HHSN268201200036C, HHSN268200800007C, HHSN268201800001C, N01HC55222, N01HC85079, N01HC85080, N01HC85081, N01HC85082, N01HC85083, N01HC85086, and grants U01HL080295 and U01HL130114 from the NHLBI, with additional contribution from the National Institute of Neurological Disorders and Stroke (NINDS). Additional support was provided by R01AG023629 from the National Institute on Aging (NIA). A full list of principal CHS investigators and institutions can be found at CHS-NHLBI.org. The content is solely the responsibility of the authors and does not necessarily represent the official views of the National Institutes of Health. Amish: The TOPMed component of the Amish Research Program was supported by NIH grants R01 HL121007, U01 HL072515, and R01 AG18728, P30 DK072488. We gratefully acknowledge our Amish liaisons and field workers and the extraordinary cooperation and support of the Amish community, without which these studies would not have been possible. ARIC: The ARIC study has been funded in whole or in part with Federal funds from the National Heart, Lung, and Blood Institute, National Institutes of Health, Department of Health and Human Services (contract numbers HHSN268201700001I, HHSN268201700002I, HHSN268201700003I, HHSN268201700004I and HHSN268201700005I). The authors thank the staff and participants of the ARIC study for their important contributions. WHI: The WHI program is funded by the National Heart,

European-ancestry participants and did not leverage the multiple observations of handgrip strength available in some studies. The genetic associations of the natural variation in handgrip strength could help understanding the biological mechanisms of aging.

We aimed to identify genetic variants (specifically rare and ancestry-specific variants) associated with handgrip strength using multi-ancestry samples and whole genome sequence data from the National Heart, Lung and Blood Institute (NHLBI) Trans-Omics in Precision Medicine (TOPMed) Program. Leveraging multi-ancestry samples can improve our understanding of a trait by identifying novel variants that are not common in all populations. For studies where multiple handgrip strength observations over time per person were available, we leveraged this information to increase the effective sample size. We compared three different methods for incorporating multiple measures of handgrip strength per person (one handgrip strength observation close to age 60 years *vs* the mean of all handgrip strength observations *vs* inclusion of all handgrip strength observations and accounting for correlations between multiple measurements). We additionally leveraged the different ancestries available in the UKBB to further explore the ancestry-specific association signals identified in TOPMed.

## Materials and methods

### Populations and participants

The study design included TOPMed participants from six cohorts who had handgrip strength measurements available: the Old Order Amish study, the Atherosclerosis Risk in Communities (ARIC) Study, the Cardiovascular Health Study (CHS), the Framingham Heart Study (FHS), the Hypertension Genetic Epidemiology Network (HyperGEN) and the Women's Health Initiative (WHI). Descriptions of each cohort are available in S1 File. These participants represented six study-reported population groups: Non-Hispanic Whites/European-Americans (EA), Non-Hispanic Blacks/African-Americans (AA), Hispanics/Latinos, Asians/Pacific-Islanders, American-Indian/Alaskan-Native, and Other.

### Handgrip strength harmonization

The number of handgrip strength measures from multiple visits per study participant varied across studies (ranging from one to nine). Details on the handgrip strength measurement protocol for each study are available in S1 File. For each exam with a handgrip strength measure, we selected the maximum value for each participant. This approach is the standard handgrip strength protocol used in many epidemiological and GWAS studies [1, 2, 18, 20]. We removed weak handgrip strength observations ($< 5$kg) that were more likely to be measurement errors than real values. In addition, for studies with handgrip strength available at multiple study visits (FHS, CHS and WHI), we removed outliers (|minimum or maximum handgrip strength—mean handgrip strength| $> 20$kg by participant). In total, we removed 123 observations (weak value or outliers). The final sample size consisted of 13,552 unique participants contributing 36,872 handgrip strength observations.

### Whole genome sequencing

The NHLBI TOPMed program provided whole genome sequence (WGS), performed at an average depth of 38× by several sequencing centers (New York Genome Center; Broad Institute of MIT and Harvard; University of Washington Northwest Genomics Center; Illumina Genomic Services; Psomagen, Inc.; and Baylor Human Genome Sequencing Center), using DNA from blood. Details regarding the laboratory methods, data processing and quality control are described in Taliun et al. [21] and in documents included in each TOPMed accession

Lung, and Blood Institute, National Institutes of Health, U.S. Department of Health and Human Services through contracts HHSN268201600018C, HHSN268201600001C, HHSN268201600002C, HHSN268201600003C, and HHSN268201600004C. HyperGEN: The HyperGEN Study is part of the NHLBI Family Blood Pressure Program; collection of the data represented here was supported by grants U01 HL054472 (MN Lab), U01 HL054473 (DCC), U01 HL054495 (AL FC), and U01 HL054509 (NC FC). The HyperGEN: Genetics of Left Ventricular Hypertrophy Study was supported by NHLBI grant R01 HL055673 with whole-genome sequencing made possible by supplement -18S1. Individual grants Dr. Jeffrey R. O'Connell grant U01 HL137181 NIH grant U24AG051129 awarded by the National Institute on Aging (NIA). UK Biobank This research has been conducted using the UK Biobank Resource under Application Number 42614. The funders had no role in study design, data collection and analysis, decision to publish, or preparation of the manuscript.

**Competing interests:** I have read the journal's policy and the authors of this manuscript have the following competing interests: Dr. Bruce Psaty serves on the Steering Committee of the Yale Open Data Access Project funded by Johnson & Johnson. This does not alter our adherence to PLOS ONE policies on sharing data and materials. The other co-authors do not have conflicts of interest to declare.

released on the database of Genotypes and Phenotypes (dbGaP). Processing of whole genome sequences was harmonized across genomic centers using a standard pipeline [22]. This study utilized genotypes from TOPMed 'freeze 8' that comprised 186K samples, although fewer actual participants were included based on the availability of handgrip strength measures. Variant discovery and genotype calling were performed jointly across TOPMed and Center for Common Disease Genetics (CCDG) studies for all samples using the GotCloud pipeline [23]. A support vector machine quality filter was trained using known variants (positive training set) and Mendelian-inconsistent variants (negative training set). The TOPMed Data Coordinating Center performed additional quality control checks for sample identity issues including pedigree errors, sex discrepancies, and genotyping concordance. The reads were aligned to human genome build GRCh38 using a common pipeline across all centers. After site level filtering, TOPMed freeze 8 consisted of 1.02B variants (single nucleotide variants (SNVs) and short insertion-deletion (Indels) variants). As the analysis methods we use require no missing genotype data, we included in our analyses genotypes with a low depth (less than 10-fold average depth) rather than imputing them to the mean, as is done by most software. The genotype files were coded using the same reference allele for all studies and ancestries and the "minor" allele was defined based on the overall allele frequency in TOPMed.

## Principal Component Analyses (PCA) and relatedness estimates

PCA and relatedness (IBD) estimates were made for 140,062 TOPMed samples using 638,486 SNVs passing QC, linkage disequilibrium (LD)-pruned from the genotypes with minimal depth >10× with a minor allele frequency (MAF) threshold of 0.01 (where "minor" allele is defined based on overall allele frequency in all of TOPMed), a missing call rate (MCR) threshold of 0.01 and a LD threshold of 0.32. The ancestral principal components (PCs) data were created by running the GENetic EStimation and Inference in Structured samples (GENESIS) R package 'pcair' function [24]. PC-AiR partitions samples into 'unrelated' and 'related' sets based on genotypes and then performs PCA on the 'unrelated' set, and finally projects PC values for the 'related' set [25]. A threshold of $2^{(-9/2)}$ as inferred by the KING ibdseg algorithm [26] was used for defining the 'unrelated' set, meaning that a set of samples less related than $3^{rd}$ degree was used to calculate the PCs. Relatedness was estimated using the GENESIS R package 'pcrelate' function [24], which estimates kinship coefficients and IBD sharing probabilities conditional on ancestry [27]. The resulting relatedness matrix includes pairwise kinship estimates for all samples, not just those used to calculate PCs. After reviewing the PCA plots as well as the PC-SNP correlation plots, we determined that PCs 1–11 detected ancestry among the TOPMed freeze 8 samples, and included PCs 1–11 in all analysis models, as recommended by the TOPMed Data Coordinating Center (for an example from an earlier TOPMed data freeze, see Extended Data Fig 1 in Taliun et al. 2021 [21]).

## Association analyses in TOPMed

Harmonized handgrip strength data were pooled across six studies and WGS association analysis of handgrip strength was performed using GENESIS on the Analysis Commons [28]. For our primary model of handgrip strength outcome (*MEAN*), we computed the mean of the maximum handgrip strength observations across exams for each individual and the mean covariate values. When height or weight information was not available at one exam, we imputed it based on the closest exams where the information was available (~1.2% of CHS participants, and ~5% of FHS participants). We used linear mixed effects models to test the association of handgrip strength with the genetic variants individually, while adjusting for age at handgrip strength measurement, age at handgrip strength measurement[2], sex, height, body

mass index (BMI), study, interaction of age, BMI and study with sex, and 11 PCs, allowing for heterogeneous variance across study groups [29]. The adjustment for study accounted for any differences due to differing study design. For studies with different mixes of ancestries, it also partially adjusted for ancestry. The inclusion of PCs in the model, in addition to study, adjusted for the residual population structure in the studies. We accounted for relatedness using a dense empirical kinship matrix. For the X chromosome, genotypes were coded as 0 and 2 for men and 0, 1 and 2 for women.

We compared our primary analysis (*MEAN*) with two additional models of handgrip strength outcome: one model where we selected handgrip strength at the visit where a participant's age was close to 60 years (*ONE*), and one model where we used all available visits when a participant had handgrip strength measures performed (*ALL*). For the latter, we conducted analyses using Generalized Linear Mixed Model Association Tests (GMMAT) [30] for variants with a MAF greater or equal to 0.001 with a random intercept to account for the correlations of handgrip strength measures from the same individuals across visits. To compare the three analysis strategies, we calculated correlations of effect sizes and P-values for each SNV from the three different models of handgrip strength. The handgrip strength phenotypic correlation between the *MEAN* and the *ONE* analysis was equal to 0.96, 0.95 and 0.93 for FHS, WHI and CHS respectively.

To estimate the impact of using multiple handgrip strength measures on the sample size, we calculated an effective sample size ($N_{eff}$) using two different methods. We first calculated variance ratios between analyses leveraging multiple observations (*ALL* or *MEAN*) versus one (*ONE*). We also calculated a participant effective sample size using: $neff = \frac{n}{1+(n-1)\times\rho}$ with $\rho$ being the study-specific mean of the maximum correlation observed across visits in men and women (as men and women had different pairwise correlations), and n being the number of handgrip strength observations for the participant. When $\rho$ approaches 1, $n_{eff}$ shrinks toward 1 and $N_{eff} = \Sigma neff$ shrinks towards the total number of participants (N). As the maximum correlation observed across exams was used, this method may have underestimated the true effective sample size.

The effect allele frequency (EAF) was estimated in the relevant group used in the analyses. The minor allele count (MAC) was derived based on the sample size (N) and the MAF (i.e., MAC = 2×N×MAF), where the MAF was defined as the EAF if EAF<0.50 and 1-EAF if EAF> = 0.50. Single-variant association analyses excluded variants with a minor allele count (MAC) less than 20. We used a significance threshold of $P < 2\times10^{-8}$ to report an association as genome-wide significant, which was slightly more stringent than the widely adopted P-value threshold of $5\times10^{-8}$ in GWAS, based on estimations for genome-wide significance for WGS studies in UK10K [31].

For the main analysis (*MEAN*), we additionally performed ancestry-specific analyses (analyses conducted separately in EA and AA, the two population groups with the largest sample sizes) and sex-stratified analyses (analyses conducted separately in men and women). The heterogeneity test between men and women was calculated using: $\frac{(\beta men-\beta women)^2}{SEmen^2+SEwomen^2-2\times r\times SEmen\times SEwomen}$ that assessed the difference in effect sizes between men ($\beta_{men}$) and women ($\beta_{women}$) while accounting for correlation (r) due to relatedness, calculated using the genome-wide correlation of the effect sizes in men and women.

We also performed gene-based tests with SKAT-O using GENESIS [32] for variants with a MAC greater or equal to 2 and a MAF lower than 0.01. We performed analyses in the pooled sample as well as stratified by ancestry, as rare variant aggregation tests can be underpowered for identifying rare variant associations in admixed populations [33]. We aggregated variants in GENECODE (v28) genes using three successively less stringent criteria: high confidence

loss of function (hcLoF) based on Loss-Of-Function Transcript Effect Estimator [https://github.com/konradjk/loftee], loss of function (LoF), or LoF and missense variants combined, based on Variant Effect Predictor (VEP) Ensembl Consequence [34]. We also tested the association of aggregated variants by a fourth strategy which enriched for likely deleterious variants in protein coding genes by retaining variants which were 1) hcLoF or 2) missense and predicted to be deleterious (by either SIFT4G [35], Polyphen2 (HDIV or HVAR) [36], or LRT [37]) or 3) for which VEP Ensembl Consequence has the terms inframe_insertion or inframe_deletion, and which have Functional Analysis through Hidden Markov Models (fathmm [38]) XF coding_score >0.5 (pathogenic mutations). The annotation based variant filtering and gene-based aggregation was performed using TOPMed freeze 8 WGSA Google BigQuery annotation database on the BiodataCatalyst powered by Seven Bridges platform [https://biodatacatalyst.nhlbi.nih.gov/]. The annotation database was built using variant annotations generated by Whole genome Sequence annotator version v0.8 [39] and formatted by WGSA-Parsr version 6.3.8 [https://github.com/UW-GAC/wgsaparsr]. We used a significance threshold of $P < 5 \times 10^{-7}$ to report a gene as genome-wide significant, which corresponded to a Bonferroni correction for the number of genes tested and the number of analyses performed. Only genes with a total MAC greater than or equal to 5 were analyzed.

## Association analyses in the UKBB

The UKBB was used as an independent sample for external validation of our findings. The UKBB is a prospective cohort study with deep genetic and phenotypic data collected on approximately 500,000 individuals from across the United Kingdom, aged between 40 and 69 at recruitment [40]. The centralized analysis of the genetic data, including genotype quality, population structure and relatedness of the genetic data, and efficient phasing and genotype imputation has been described extensively elsewhere [40]. Two similar arrays were used for genotyping (Applied Biosystems UK Biobank Lung Exome Variant Evaluation and UK Biobank Axiom Arrays) and pre-phasing was performed using markers present on both arrays. Phasing on the autosomes was carried out using SHAPEIT3 [41] and 1000 Genomes phase 3 panel to help with the phasing of non-European ancestry samples. Imputations were carried out using the IMPUTE4 program [40] with the Haplotype Reference Consortium (HRC) reference panel or with a merged UK10K and 1000 Genomes phase 3 reference panel when a variant was not present in HRC. For chromosome X, haplotype estimation and genotype imputation were carried out separately on the pseudo-autosomal and non-pseudo autosomal regions. The top 40 ancestral PCs were generated using fastPCA [42], a set of 407,219 unrelated participants with high quality samples and 147,604 high quality markers pruned to minimize LD. The corresponding PCs-loadings were then computed, and all samples were projected onto the PCs, thus forming a set of PC scores for all samples in the cohort.

UKBB African-ancestry participants were identified using the following six self-reported ancestries: "Caribbean", "African", "Black or Black British", "Any other Black background", "White and Black African" and "White and Black Caribbean". We identified UKBB European-ancestry participants using the following three self-reported ancestries: "White", "British", and "Irish". Handgrip strength in the UKBB was measured at three exams but the majority of the participants had handgrip strength measured at only the first exam so we used only the first exam. We excluded participants with a weak handgrip strength (< 5kg), and samples with high heterozygosity and high missing rate, sex aneuploidy, and with different genomic and stated sex. We used the Scalable and Accurate Implementation of Generalized mixed model (SAIGE) [43] and linear mixed effect models to evaluate the variants that were associated with handgrip strength in TOPMed, adjusting for age at handgrip strength measurement, age at

handgrip strength measurement[2], sex, height, BMI, age×sex, BMI×sex, and PCs significantly associated with handgrip strength, and using an empirical kinship matrix.

## Evaluation of previously published handgrip strength GWAS signals

We performed a look-up of the 1,452 SNVs that passed the genome-wide threshold in the UKBB handgrip strength GWAS (Stage 1) [18] in our TOPMed analyses and calculated the correlations of the effect sizes and frequencies of the effect alleles between the UKBB handgrip strength GWAS and the TOPMed WGS.

## Assessment of expression or methylation quantitative trait loci

We investigated whether the genetic variants from the handgrip strength UKBB GWAS or the primary *MEAN* TOPMed handgrip strength WGS were expression or methylation quantitative trait loci (eQTLs / meQTLs) in human skeletal muscle using publicly available results from the FUSION study [44].

## Ethic statement

The data across the following studies were shared through the database of Genotype and Phenotype (dbGaP) exchange area. The study was approved by the appropriate institutional review boards (IRB) and informed consent was obtained from all participants. Amish: All study protocols were approved by the IRB at the University of Maryland Baltimore. Informed consent was obtained from each study participant. ARIC: The ARIC Study has been approved by IRB at all participating institutions: University of North Carolina at Chapel Hill, Johns Hopkins University, University of Minnesota, and University of Mississippi Medical Center. Study participants provided written informed consent at all study visits. CHS: All CHS participants provided informed consent, and the study was approved by the IRB or ethics review committee of University Washington. FHS: The Framingham Heart Study was approved by the IRB of the Boston University Medical Center. All study participants provided written informed consent. HyperGEN: All HyperGEN participants provided informed consent, and the study was approved by the IRB of the University of Kentucky. WHI: All WHI participants provided informed consent and the study was approved by the IRB of the Fred Hutchinson Cancer Research Center.

# Results

## Populations and participants

WGS association analysis of handgrip strength included 13,552 TOPMed participants. The majority of these participants belonged to two major population groups: EA (N = 10,263; 76%) and AA (N = 3,145; 23%) (**Table 1**). Around 1% of the sample belonged to other population groups (Hispanic or Latino, Asian or Pacific-Islander, American-Indian or Alaskan-Native, or other). Additional descriptive tables of the participants are presented in **S1–S4 Tables in S1 File**. Mean age (SD) of participants was 69 years (14) and mean handgrip strength (SD) was 29.4 kg (11.2). A total of 4,878 participants (36%) were men. Mean handgrip strength (SD) was equal to 23.5 kg (6.5) in women and 40.1 kg (9.9) in men. Mean age (SD) was 71 years (11) in EA participants and 60 years (18) in AA participants. Mean handgrip strength was equal to 28.9 kg (10.9) in EA participants and 31.6 kg (11.9) in AA participants.

**Table 1. Description of the 13,552 TOPMed participants included in the main pooled whole genome sequence association analysis of MEAN handgrip strength and the 8,408 UK Biobank participants used for replication.**

| | TOPMED | | | | | | UKBB AA[a] |
|---|---|---|---|---|---|---|---|
| | TOTAL | EA[a] | AA[a] | Other[a] | MEN | WOMEN | TOTAL |
| N (%) | 13,552 (100) | 10,263 (76) | 3,145 (23) | 144 (1) | 4,878 (36) | 8,674 (64) | 8,408 (100) |
| Age (years), mean (SD) | 69 (14) | 71 (11) | 60 (18) | 76 (6) | 68 (14) | 69 (14) | 52 (8) |
| Height (cm), mean (SD) | 165 (9.4) | 165 (9.5) | 166 (9.3) | 158 (6.5) | 174 (6.9) | 160 (6.5) | 168 (9) |
| BMI (kg/m2), mean (SD) | 28.4 (5.8) | 27.7 (5.2) | 30.9 (7.0) | 27.6 (5.3) | 28.2 (4.8) | 28.6 (6.3) | 29.3 (5.4) |
| Handgrip strength (kg), mean (SD) | 29.4 (11.2) | 28.9 (10.9) | 31.6 (11.9) | 21.4 (7.2) | 40.1 (9.9) | 23.5 (6.5) | 34.0 (11.7) |

Mean covariates were calculated across all exams for which handgrip strength was available for the *MEAN* analysis.

[a]EA: White/European-American; AA: Black/African-American/African-ancestry; Other: Hispanic or Latino (N = 107), Asian or Pacific-Islander (N = 17), American-Indian or Alaskan-Native (N = 8), or other (N = 12).

[b] We identified UKBB African-ancestry participants using the following six self-reported ancestries: "Caribbean", "African", "Black or Black British", "Any other Black background", "White and Black African" and "White and Black Caribbean".

## Main single-variant analyses results

Manhattan plots and Quantile-Quantile plots for the main analysis (*MEAN*) are included in **S1 and S2 Figs in S1 File**. We observed a slight inflation of type I error ($\lambda_{GC}$ = 1.08), despite having adjusted for potential sources of heterogeneity, cryptic relatedness, and population structure. This genomic inflation factor, computed at the median, was not extreme and within the range of many published GWAS of densely typed variants with mixed ancestries. We detected rare or low-frequency signals in ten regions at the genome-wide significance threshold ($P < 2 \times 10^{-8}$) in at least one analysis (pooled sample of 13,552 participants, ancestry-specific, or sex-specific analyses for the *MEAN* handgrip strength outcome model). The most significant variant in each region is indicated in **Table 2** and **S5 Table in S1 File**. All signals were specific to AA, the variants being too rare in EA to be tested in that subgroup alone. We identified a region in 8p12 with a variant (rs2958754) common in AA (MAF = 0.10) but rare in EA (MAF = 0.001), and with the highest association in women (**S5 Table in S1 File**, $P = 1.3 \times 10^{-8}$).

**Table 2. Main association results from the WGS association analysis of *MEAN* handgrip strength ($P < 2 \times 10^{-8}$ in at least one analysis) in the pooled sample of 13,552 TOPMed participants, or stratified by study-reported population groups (AA), and by sex (AA men and AA women).**

| Chr | Pos (build 38) | rsid | Ref | Alt | TOTAL (N = 13,552) | | | AA[a] (N = 3,145) | | | AA MEN (N = 987) | | | AA WOMEN (N = 2,158) | | | Gene |
|---|---|---|---|---|---|---|---|---|---|---|---|---|---|---|---|---|---|
| | | | | | EAF | Beta | P | EAF | Beta | P | EAF | Beta | P | EAF | Beta | P | |
| 2 | 198,070,149 | rs74688411 | A | G | 0.004 | 3.84 | 6.4E-09 | 0.018 | 3.95 | 7.5E-08 | 0.018 | 6.21 | 2.5E-04 | 0.018 | 2.55 | 4.0E-04 | *PLCL1* |
| 5 | 76,919,524 | rs57776684[b] | C | T | 0.003 | -2.84 | 5.1E-04 | 0.013 | -3.14 | 3.8E-04 | 0.011 | -13.10 | 1.5E-08 | 0.014 | -0.51 | 5.4E-01 | *S100Z* |
| 5 | 174,630,802 | rs377692678 | G | A | 0.0008 | -8.66 | 8.0E-09 | 0.003 | -7.63 | 3.5E-06 | 0.006 | -8.98 | 3.2E-03 | 0.002 | -6.03 | 4.0E-03 | *intergenic* |
| 7 | 95,524,137 | rs544430450 | G | A | 0.001 | -6.66 | 1.7E-08 | 0.006 | -6.10 | 2.2E-06 | 0.006 | -6.00 | 4.1E-02 | 0.006 | -5.62 | 1.9E-05 | *ASB4* |
| 8 | 30,273,968 | rs2958754[b] | G | A | 0.023 | -1.28 | 1.0E-05 | 0.096 | -1.27 | 1.1E-04 | 0.085 | -0.19 | 8.2E-01 | 0.101 | -1.55 | 1.7E-06 | *intergenic* |
| 10 | 85,774,936 | rs569475444[b] | G | C | 0.005 | -1.76 | 5.5E-03 | 0.022 | -1.80 | 9.1E-03 | 0.024 | -8.16 | 9.4E-08 | 0.021 | 0.97 | 1.6E-01 | *GRID1* |
| 10 | 119,692,415 | rs189542078[b] | C | T | 0.001 | 5.36 | 1.0E-05 | 0.005 | 5.16 | 1.0E-04 | 0.005 | -1.08 | 7.5E-01 | 0.006 | 6.29 | 8.3E-07 | *intergenic* |
| 11 | 113,977,732 | rs182799368[b] | G | T | 0.003 | -3.13 | 4.2E-05 | 0.013 | -3.07 | 2.3E-04 | 0.016 | -9.19 | 2.0E-07 | 0.012 | 0.06 | 9.4E-01 | *HTR3A* |
| 14 | 96,043,941 | rs143569685[b] | T | A | 0.003 | -3.57 | 8.1E-07 | 0.014 | -3.57 | 8.0E-06 | 0.013 | -2.31 | 2.4E-01 | 0.015 | -3.93 | 4.3E-07 | *C14orf132* |
| 18 | 71,404,509 | rs185725127 | A | C | 0.001 | -6.92 | 1.4E-09 | 0.006 | -6.52 | 1.9E-07 | 0.006 | -7.21 | 1.6E-02 | 0.006 | -5.97 | 2.6E-06 | *intergenic* |

EAF: Effect allele Frequency; Alt: Alternate (Effect) allele; Pos: Positions in build GRCh38; Beta: effect size, unit (kg).

[a]AA: Black/African American/African-ancestry participants.

[b] SNP significant in sex-stratified analyses (**S5 Table in S1 File**).

We identified sex-specific signals ($P_{het}$<0.05) in six regions (**S5 Table in S1 File**). The most heterogeneous associations in 5q13, 10q23 and 11q23 were genome-wide significant in men but not significant in women.

## Gene-based tests results

We detected one gene (*SLCO1A2*) associated with handgrip strength at the genome-wide significance level ($P < 5 \times 10^{-7}$) when aggregating hcLoF/LoF rare variants with a MAC $\geq$ 2 and a MAF <1% using SKAT-O. The association was EA-specific (**Table 3**). Among the six hcLoF variants that contributed to the test, two frameshift mutations were associated with handgrip strength in single-variant association analyses (12:21292255:AAC:A, delAC, rs777190986, MAF = 0.0001, MAC = 3, $P = 8.6 \times 10^{-4}$ and 12:21306938:CTGTT:C, delTGTT, rs761787824, MAF = 0.0002, MAC = 4, $P = 2.6 \times 10^{-6}$). Additionally, a total of four genes were suggestively associated with handgrip strength ($P < 10^{-5}$) in at least one analysis (all participants, EA only, or AA only) when aggregating 1) hcLoF, 2) LoF, or 3) hcLoF, missense variants predicted to be deleterious, and pathogenic indels (**Table 3**). One EA-specific association (*C9orf43*) was found when aggregating hcLoF or LoF and was driven by a splice donor mutation (rs766369889, MAF = 0.0002, MAC = 3, $P = 1.4 \times 10^{-6}$). Two EA-specific associations (*ZNF593* and *GALNT17*) were found when aggregating hcLoF, missense variants predicted to be deleterious, and pathogenic indels. The first association (*ZNF593*) was driven by a missense variant (rs2232649, MAF = 0.004, MAC = 112, $P = 1.3 \times 10^{-5}$). This variant has a MAF higher than 1% in AA and thus it did not contribute to the gene-based test in AA only. The second association (*GALNT17*) was driven by a missense variant (rs139969494, MAF = 0.001, MAC = 22, $P = 1.5 \times 10^{-6}$). One AA-specific association (*MRPL16*) was found when aggregating hcLoF,

**Table 3. Significant and suggestive genes (P<10⁻⁵) associated with *MEAN* handgrip strength in TOPMed when aggregating rare variants with a minor allele count (MAC) greater or equal to 2 and a minor allele frequency (MAF) less than 1% using SKAT-O.**

| Gene | Group[a] | High confidence loss of function (hcLoF) variants | | | | LoF variants | | | | LoF and missense variants | | | | hcLoF, deleterious missense variants and pathogenic indels | | | |
|---|---|---|---|---|---|---|---|---|---|---|---|---|---|---|---|---|---|
| | | n.site | n.alt | n.s.alt | *P* | n.site | n.alt | n.s.alt | *P* | n.site | n.alt | n.s.alt | *P* | n.site | n.alt | n.s.alt | *P* |
| *ZNF593* | Total | -- | -- | -- | -- | -- | -- | -- | -- | 11 | 343 | 335 | 0.001 | 6 | 129 | 129 | 4.7E-06 |
| | EA | -- | -- | -- | -- | -- | -- | -- | -- | 11 | 47 | 44 | 0.07 | 6 | 20 | 20 | 0.01 |
| | AA | -- | -- | -- | -- | -- | -- | -- | -- | -- | -- | -- | -- | -- | -- | -- | -- |
| *MRPL16* | Total | 2 | 10 | 10 | 0.24 | 3 | 12 | 12 | 0.11 | 17 | 250 | 239 | 0.75 | 13 | 210 | 201 | 0.65 |
| | EA | 2 | 9 | 9 | 0.15 | 3 | 11 | 11 | 0.06 | 14 | 92 | 90 | 0.34 | 11 | 86 | 84 | 0.36 |
| | AA | -- | -- | -- | -- | -- | -- | -- | -- | 5 | 80 | 79 | 0.004 | 3 | 24 | 24 | 1.2E-06 |
| *SLCO1A2* | Total | 6 | 18 | 18 | 2.7E-07 | 6 | 18 | 18 | 2.7E-07 | 47 | 434 | 419 | 0.37 | 31 | 250 | 240 | 0.37 |
| | EA | 4 | 10 | 10 | 1.3E-06 | 4 | 10 | 10 | 1.3E-06 | 31 | 148 | 148 | 0.12 | 20 | 77 | 77 | 0.02 |
| | AA | 2 | 7 | 7 | 0.81 | 2 | 7 | 7 | 0.81 | 15 | 80 | 73 | 0.30 | 10 | 34 | 27 | 0.42 |
| *GALNT17* | Total | -- | -- | -- | -- | -- | -- | -- | -- | 19 | 279 | 278 | 0.03 | 14 | 98 | 98 | 3.1E-05 |
| | EA | -- | -- | -- | -- | -- | -- | -- | -- | 12 | 81 | 81 | 1.1E-05 | 10 | 74 | 74 | 9.3E-06 |
| | AA | -- | -- | -- | -- | -- | -- | -- | -- | 5 | 69 | 69 | 0.76 | 4 | 21 | 21 | 0.42 |
| *C9orf43* | Total | 3 | 9 | 9 | 1.0E-05 | 4 | 12 | 12 | 1.5E-05 | 28 | 409 | 404 | 0.21 | 14 | 63 | 63 | 0.05 |
| | EA | 2 | 5 | 5 | 9.4E-06 | 2 | 5 | 5 | 9.4E-06 | 17 | 315 | 311 | 0.50 | 10 | 49 | 49 | 0.06 |
| | AA | -- | -- | -- | -- | 2 | 7 | 7 | 0.46 | 13 | 131 | 130 | 0.80 | 4 | 30 | 30 | 0.46 |

n.site: the number of variant sites included in the test; n.alt: the number of alternate alleles included in the test; n.s.alt: the number of samples with an observed alternate allele at any variant in the aggregate set.

–: no result (total minor allele count lower than 5).

[a]EA: White/European-American; AA: Black/African-American/African-ancestry.

missense variants predicted to be deleterious, and pathogenic indels and was driven by a missense variant predicted to be deleterious by SIFT (rs147545257, MAF = 0.002, MAC = 15, $P = 6.8 \times 10^{-7}$).

## Comparison of handgrip strength outcome models and calculation of the effective sample size

We observed very high correlations when comparing effect-sizes of variants with a MAF greater or equal to 0.001 from the three different models of handgrip strength outcomes (r ranging from 0.93 to 0.99). The correlations when comparing–log(P)-values were also high but a bit lower, particularly when comparing the analysis leveraging one handgrip strength observation (*ONE*) *versus* the analyses leveraging the multiple handgrip strength observations (*MEAN* and *ALL*, r = 0.87 and 0.85 respectively). The highest correlation (r = 0.98 for–log(P)-values and r = 0.99 for effect sizes) was observed when comparing *ALL* and *MEAN*, the two models that leveraged the multiple handgrip strength observations (**S3 and S4 Figs in S1 File**). Results of all signals reaching genome-wide significance with at least one model of handgrip strength outcome (*ALL*, *ONE* and *MEAN*) and a MAF greater or equal to 0.001 are presented in **S6 Table in S1 File**. All these signals had a P-value lower than $10^{-7}$ in the *MEAN* analysis. We estimated that the effective sample size increase ranged between 7 and 12% when leveraging multiple handgrip strength observations and accounting for the correlations of the observations (**S7 Table in S1 File**).

## Exploration of the main ancestry-specific TOPMed results in UKBB

We did not observe a significant association of the six low-frequency or common index variants associated with *MEAN* handgrip strength in TOPMed AA association analysis in UKBB African-ancestry participants (N = 8,408; **S8 Table in S1 File**). Two of the six variants (rs577776684 and rs569475444), which had a low-frequency in TOPMed AA, were rarer in the UKBB African-ancestry participants.

## Evaluation of previously published handgrip strength GWAS signals

Single-variant associations for the 16 index variants from the CHARGE+UKBB handgrip strength GWAS [18] are presented in **S9 Table in S1 File**. We estimated that the power to detect each of these 16 variants in our TOPMed sample, using significance level α = 0.05, ranged between 0.18 and 0.26, assuming that the variants explain the same proportion of variance in our sample as in the GWAS discovery UKBB sample. In addition, a total of 1,452 SNVs were genome-wide significant in the handgrip strength GWAS Stage 1 (UKBB only). We observed a high concordance of the direction of effects between UKBB and TOPMed (98% when including all participants in the TOPMed analysis, 99% when including only TOPMed EA, and 92% when including only TOPMed AA). We observed a high concordance of the effect sizes reported by the UKBB handgrip strength GWAS and TOPMed (correlation of 0.93 when including all TOPMed participants in the analysis or only EA, and of 0.69 when including only TOPMed AA in the analysis). We observed a difference in allele frequencies between EA and AA for the handgrip strength UKBB GWAS results in TOPMed (**S5 Fig in S1 File**). A total of 295 variants showed nominal significance (*P*<0.05) and 14 variants had a P-value less than 0.001 in at least one TOPMed analysis (all participants, ancestry-specific, or sex-specific analyses for the *MEAN* handgrip strength outcome model) (**S10 Table in S1 File**). The most significant association was observed for the variant rs4793937 in 17q21 ($P = 3.8 \times 10^{-6}$) with a higher effect in men compared to women.

## Assessment of expression or methylation quantitative trait loci

We investigated whether the 16 lead SNPs from the handgrip strength UKBB GWAS, the 14 SNPs from the handgrip strength UKBB GWAS with a P-value less than 0.001 in TOPMed, and the 10 lead SNPs from the TOPMed handgrip strength WGS, were eQTLs or meQTLs in human skeletal muscle. We found that 13 SNPs from the handgrip strength UKBB GWAS were eQTLs ($P<1\times10^{-6}$) in 1p36 (*PEX14*), 2p21 (*LRPPRC*), 17q21 (*HOXB* cluster) and 17q25 (*ACTG1*). Interestingly, the allele associated with a higher handgrip strength was also associated with an increase in gene expression, except for rs6565586 in 17q25 (**S11 Table in S4 File**). We found that all of these eQTLs were also meQTLs ($P<1\times10^{-6}$) except rs4245797 and we identified additional meQTLs in 1p32, 2p22, 2p13, 10q24, 10q26, 11q25, and 12p12 (**S12 Table in S4 File**).

## Discussion

This study is the first to report a WGS association analysis of handgrip strength in 13,552 participants from six studies representing diverse population groups. By leveraging multiple handgrip strength observations per person over time, we increased our effective sample size by 7–12%. Using single-variant analyses, we identified 10 new loci influencing handgrip strength with rare, low-frequency or common AA-specific associations. Using gene-based tests, we identified one significant and four suggestive genes associated with handgrip strength when aggregating rare and functional variants. These associations were ancestry specific.

Several genetic variants identified associated with handgrip strength using single-variant association analysis or gene-based tests in TOPMed lie in genes with a function relevant to the brain or muscle. We describe below the function of some of these genes.

Like the findings in Willems et al handgrip strength GWAS, two TOPMed handgrip strength hits lie within genes with a function relevant to the brain. The genetic variant rs569475444 lies in the intron of the glutamate ionotropic receptor delta type subunit 1 gene (*GRID1*) that encodes a subunit of glutamate receptor channels and mediates most of the fast excitatory synaptic transmission in the central nervous system and play key roles in synaptic plasticity. It is mainly expressed in the brain (Genotype-Tissue Expression project, GTEx) and sex-specific associations in this gene were reported with Alzheimer's Disease risk [45]. Additionally, sex differences in the glutamate system have been described [46]. The genetic variant rs182799368 lies in the intron of the 5-hydroxytryptamine receptor 3A gene (*HTR3A*) that encodes the subunit A of the type 3 receptor for 5-hydroxytryptamine (serotonin), a biogenic hormone that functions as a neurotransmitter, a hormone, and a mitogen. Variants in this gene were reported to be associated with multiple sclerosis [47], a potentially disabling disease of the brain and spinal cord causing numbness or weakness in one or more limbs, tremor, lack of coordination or unsteady gait. Additionally, sex differences in serotonin synthesis rates have been described in human brain [48]. Emerging evidence suggests that handgrip strength is associated with better cognitive performance in patients with major depressive disorder (MDD) and that stronger handgrip strength is associated with greater left and right hippocampal volume and reduced white matter hyperintensities in patients with MDD [14]. White matter hyperintensities are common, albeit mild, in middle adult life and are associated with physical disability, possibly through reduced speed, fine motor coordination, and muscular strength [49]. A correlation also exists between spinal white matter organization as determined by diffusion tensor imaging and force control in precision grip, an important aspect of manual dexterity in healthy subjects [50].

Other TOPMed handgrip strength hits lie within genes with a function relevant to muscle or have reported associations with muscle or age-related traits.

The genetic variant rs74688411 lies in the intron of the phospholipase C like 1 (*PLCL1*) gene that has been reported by GWAS of adult and juvenile dermatomyositis [51], hip bone size variation in women [52], and total bone mineral density [53]. Hip bone size and bone mineral density are key measurable risk factors for low trauma hip fractures. Handgrip strength is also a potentially useful objective parameter to predict fracture since it is an indicator of general muscle strength, is associated with fragility and propensity to fall, and is related to fracture risk [54, 55]. Willems et al found genome-wide genetic correlations of bone mineral density with handgrip strength, supportive of a role for genetically predicted handgrip strength in fracture risk [18]. The genetic variant rs544430450 lies in the intron of the ankyrin repeat and SOCS box containing 4 gene (*ASB4*) that colocalizes with the insulin receptor substrate 4 (IRS4) in the hypothalamic neurons and mediates IRS4 degradation [56]. Ankyrin repeat domain containing proteins have a role in muscle physiology [57]. Indeed, many aspects of muscle function are controlled by the superfamily ankyrin repeat domain containing proteins, including structural fixation of the contractile apparatus to the muscle membrane by ankyrins, the archetypical member of the family. Interestingly, this gene is highly and specifically expressed in the adrenal gland (GTEx). Variants in the solute carrier organic anion transporter family member 1A2 (*SLCO1A2*) have been reported to be associated with statin-induced myopathy and progressive supranuclear palsy, an uncommon brain disorder that affects, among others, movement, control of walking (gait) and balance. This gene is specifically expressed in the brain (GTEx). The S100 calcium binding protein Z (S100Z) and the mitochondrial ribosomal protein L16 (*MRPL16*) genes were found to be significantly different between non-frail and frail middle-aged individuals [58]. Variants in the polypeptide N-acetyl-galactosaminyltransferase 17 gene (*GALNT17*) were reported associated with physical activity [59].

We have compared our primary analysis, where all available handgrip strength measures were averaged for each participant (*MEAN*), with two additional models: one model where the handgrip strength measure at the exam close to 60 years old was analyzed for each participant (*ONE*), and one model where all available handgrip strength measures were analyzed for each participant (*ALL*). Each of these approaches has strengths and limitations. The *ONE* analysis does not leverage the multiple handgrip strength measures and requires selecting one handgrip strength exam. We decided to select the exam close to 60 years old for each participant to investigate handgrip strength in older adults at a time when handgrip strength may be declining. Thus, this approach has a limited power. The *MEAN* analysis accounts for the variability of handgrip strength measures across studies but does not account for the variability of handgrip strength measures within studies. A weighted analysis could be performed to account for the number of measures for each participant within each study, but this approach is not implemented in GENESIS or GMMAT software available on the Analysis Commons. We do not expect our results to drastically change when weights are incorporated in the model. The *ALL* analysis may be the best model to leverage all available handgrip strength measures. However, this approach is computationally intensive and may over-represent participants or studies with a higher number of handgrip strength observations (such as CHS or WHI). To note, all signals with a MAF greater or equal to 0.001 and reaching genome-wide significance ($P<2\times10^{-8}$) with at least one model of handgrip strength outcome had a P-value lower than $10^{-7}$ in the *MEAN* analysis (**S6 Table in S1 File**).

None of our TOPMed findings were found to be eQTLs or meQTLs in human skeletal muscle. However, the FUSION eQTL and meQTL study was performed in Finnish participants whereas our signals were almost all specific to AA. Availability of eQTLs or meQTLs data in African-ancestry participants is limited.

Strengths of this investigation include the analysis of extremely high-quality deep sequence data from a large, multi-ancestry sample with multiple handgrip strength observations. We included six studies with the majority of participants identified as Non-Hispanic Black/African-American or Non-Hispanic White/European-American. The high-quality sequence data allowed us to assess association of handgrip strength with genetic variation across the full allelic spectrum.

We also recognize limitations. Our sample size, and thus our power, is limited compared to previous large European-ancestry GWAS, but the WGS permits the investigation of less common and rare variants. Including larger samples of individuals, especially of non-European population groups, will improve the characterization of the genetic architecture underlying handgrip strength. We used a slightly more stringent threshold to report genome-wide significant findings than the widely adopted GWAS P-value threshold of $5 \times 10^{-8}$, to account for the higher number of variants tested (SNVs and insertions/deletions) but this threshold may still be a bit anti-conservative due to the additional subgroup analyses performed. We view these additional analyses as critical to understanding the observed associations. Finally, we have not considered in this paper local ancestry analysis that could help to further investigate the effects of the detected variants and ancestry-gene interaction that could boost statistical power of rare variant association mapping in admixed populations.

The handgrip strength genome-wide associations in single-variant analyses that we observed were AA-specific and some were also sex-specific. We observed associations with consistent direction of effects in at least two studies for each signal (**S13 Table in S1 File**). Given the large mean handgrip strength difference between men and women, we might expect that handgrip strength may have at least some different genetic associations in men and women. A recent GWAS of testosterone identified, for instance, extreme sex differences in testosterone genetic architecture [60].

Finding replication samples with ethnically diverse populations, handgrip strength information and sequence data is challenging. As no similar sample with WGS exists, we used UKBB imputed genotype data to further explore our ancestry-specific findings. We could only look-up the common or low-frequency AA-specific TOPMed signals in the UKBB African-ancestry participants, who were defined based on six different self-reported ancestries. We did not observe significant associations of these variants with handgrip strength. There are several possible explanations, including 1) genetic heterogeneity across the two samples, 2) age differences in the two samples (UKBB participants are 8 years younger on average than those in our TOPMed sample, **Table 1**) 3) imputation in UKBB African ancestry samples may not be accurate for these low frequency variants, and 4) the effects we observed may be inflated due to winners curse, and thus the power to detect the associations is limited in the UKBB African ancestry sample. Because we do not have a sequenced replication sample from the same populations as our discovery sample, we cannot rule out the possibility that the associations identified in TOPMed could be false positives.

In conclusion, by leveraging multiple handgrip strength observations from a multi-ancestry sample with sequence data, our study identified 11 new loci associated with handgrip strength with rare and/or ancestry-specific genetic variations. Further studies in ethnically diverse populations are needed to confirm these findings.

## Supporting information

**S1 File. Materials.**
(DOCX)

**S2 File. Longevity wkgp list.**
(XLSX)

**S3 File. TOPMed banner authorship.**
(XLSX)

**S4 File. S11 and S12 Tables.**
(XLSX)

## Acknowledgments

TOPMed Longevity and Healthy Aging Working Group

Co-conveners: Joanne M Murabito, Kathryn L Lunetta, and Chloé Sarnowski.

Working Group Members: Arnett, Donna K, University of Kentucky; Aviv, Abraham, Rutgers University; Baccarelli, Andrea, Columbia University; Barnes, Kathleen, University of Colorado Anschutz Medical Campus; Beame, David, University of Washington; Biggs, Mary Lou, University of Washington; Bowers, Michael, University of Washington; Bressler, Jan, University of Texas Health at Houston; Cade, Brian, Brigham & Women's Hospital; Correa, Adolfo, University of Mississippi; Dean, Jennifer, University of Virginia; DeMeo, Dawn, Brigham & Women's Hospital; Du, Margaret Mengmeng, Memorial Sloan Kettering Cancer Center; Fardo, David W, University of Kentucky; Gao, Xu, Columbia University; Gonzalez, Gisselle, Brigham & Women's Hospital; Haring, Bernhard, Medical University of Graz; Hernandez, Ryan, University of California, San Francisco; Horvath, Steve, University of California, Los Angeles; Hou, Lifang, Northwestern University; Jiang, Jicai, University of Maryland; Johnson, Craig, University of Washington; Karasik, David, Harvard University; Katsumata, Yuriko, University of Kentucky; Keely, Addison, University of Washington; Kiel, Douglas P, Harvard University; Kooperberg, Charles, Fred Hutchinson Cancer Research Center; Lange, Leslie, University of Colorado at Denver; Lira, Olivia, University of Colorado Anschutz Medical Campus; Liu, Simin, Brown University; Lunetta, Kathryn L, Boston University; Mainous, Arch (Chip) G, University of Florida; Martin, Alexandra, University of Kentucky; McGarvey, Stephen, Brown University; Mikulla, Julie, National Heart, Lung, and Blood Institute, National Institutes of Health; Murabito, Joanne M, Boston University; Musani, Solomon, University of Mississippi; Nan, Hongmei, Indiana University; Nannini, Drew, Northwestern University; O'Connell, Jeffrey R, University of Maryland; Psaty, Bruce M, University of Washington; Purnell, Jennifer Anne, University of Washington; Qiao, Dandi, Brigham & Women's Hospital; Raffield, Laura, University of North Carolina; Redline, Susan, Brigham & Women's Hospital; Regan, Elizabeth, National Jewish Health; Rich, Stephen, University of Virginia; Rotter, Jerome, Lundquist Institute; Ryan, Kathleen A, University of Maryland; Sarnowski, Chloé, Boston University; Shade, Lincoln, University of Kentucky; Silverman, Edwin, Brigham & Women's Hospital; Smith, Albert Vernon, University of Michigan; Smith, Jennifer, University of Michigan; Smoller, Sylvia, Albert Einstein College of Medicine; Thorington, Daune, Lundquist Institute; Walsh, Ann, National Heart, Lung, and Blood Institute, National Institutes of Health; Wehr, Kate, University of Washington; Yang, Ivana, University of Colorado at Denver; Zhao, Wei, University of Michigan; Zheng, Yinan, Northwestern University.

Study-specific

FHS: The FHS acknowledges the dedication of the FHS study participants without whom this research would not be possible.

CHS: A full list of principal CHS investigators and institutions can be found at CHS-NHLBI.org. The content is solely the responsibility of the authors and does not necessarily represent the official views of the National Institutes of Health.

Amish: We gratefully acknowledge our Amish liaisons and field workers and the extraordinary cooperation and support of the Amish community, without which these studies would not have been possible.

ARIC: The authors thank the staff and participants of the ARIC study for their important contributions.

## Author Contributions

**Conceptualization:** David Karasik, Douglas P. Kiel, Joanne M. Murabito, Kathryn L. Lunetta.

**Data curation:** Chloé Sarnowski, B. Gwen Windham, Joanne M. Murabito, Kathryn L. Lunetta.

**Formal analysis:** Chloé Sarnowski, Mary L. Biggs, Sylvia Wassertheil-Smoller, Jan Bressler, Marguerite R. Irvin, Kathleen A. Ryan.

**Funding acquisition:** Donna K. Arnett, L. Adrienne Cupples, Hyun Min Kang, Charles Kooperberg, Braxton D. Mitchell, Alanna C. Morrison, Bruce M. Psaty, Albert V. Smith, Ramachandran S. Vasan.

**Methodology:** Chloé Sarnowski, Han Chen, Mary L. Biggs, Sylvia Wassertheil-Smoller, Jan Bressler, Marguerite R. Irvin, Kathleen A. Ryan, David Karasik, David W. Fardo, Stephanie M. Gogarten, Benjamin D. Heavner, Deepti Jain, Arch G. Mainous, Alanna C. Morrison, Jeffrey R. O'Connell, Kenneth Rice, B. Gwen Windham, Douglas P. Kiel, Joanne M. Murabito, Kathryn L. Lunetta.

**Project administration:** Donna K. Arnett, L. Adrienne Cupples, Hyun Min Kang, Charles Kooperberg, Braxton D. Mitchell, Alanna C. Morrison, Bruce M. Psaty, Kenneth Rice, Albert V. Smith, Ramachandran S. Vasan.

**Resources:** Donna K. Arnett, L. Adrienne Cupples, Stephanie M. Gogarten, Benjamin D. Heavner, Deepti Jain, Hyun Min Kang, Charles Kooperberg, Braxton D. Mitchell, Alanna C. Morrison, Jeffrey R. O'Connell, Bruce M. Psaty, Kenneth Rice, Albert V. Smith, Ramachandran S. Vasan.

**Software:** Chloé Sarnowski, Han Chen, Stephanie M. Gogarten, Benjamin D. Heavner, Deepti Jain, Jeffrey R. O'Connell.

**Supervision:** David Karasik, Douglas P. Kiel, Joanne M. Murabito, Kathryn L. Lunetta.

**Writing – original draft:** Chloé Sarnowski, Joanne M. Murabito, Kathryn L. Lunetta.

**Writing – review & editing:** Chloé Sarnowski, Han Chen, Mary L. Biggs, Sylvia Wassertheil-Smoller, Jan Bressler, Marguerite R. Irvin, Kathleen A. Ryan, David Karasik, Donna K. Arnett, L. Adrienne Cupples, David W. Fardo, Stephanie M. Gogarten, Benjamin D. Heavner, Deepti Jain, Hyun Min Kang, Charles Kooperberg, Arch G. Mainous, Braxton D. Mitchell, Alanna C. Morrison, Jeffrey R. O'Connell, Bruce M. Psaty, Kenneth Rice, Albert V. Smith, Ramachandran S. Vasan, B. Gwen Windham, Douglas P. Kiel, Joanne M. Murabito, Kathryn L. Lunetta.

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
