## [Decision Letter · Decision Letter 0]

29 Mar 2021

PONE-D-20-39658

Identification of novel and rare variants associated with handgrip strength using whole genome sequence data from the NHLBI Trans-Omics in Precision Medicine (TOPMed) Program

PLOS ONE

Dear Dr. Sarnowski,

Thank you for submitting your manuscript to PLOS ONE. After careful consideration, we feel that it has merit but does not fully meet PLOS ONE’s publication criteria as it currently stands. Therefore, we invite you to submit a revised version of the manuscript that addresses the points raised during the review process.

We look forward to receiving your revised manuscript.

Kind regards,

Heming Wang, PhD

Academic Editor

PLOS ONE

Journal Requirements:

'I have read the journal's policy and the authors of this manuscript have the following competing interests: Dr. Bruce Psaty serves on the Steering Committee of the Yale Open Data Access Project funded by Johnson & Johnson. The other co-authors do not have conflicts of interest to declare.

5. One of the noted authors is a group or consortium (TOPMed Longevity and Healthy Aging Working

Group and the NHLBI Trans-Omics for Precision Medicine (TOPMed) Consortium). In addition to naming the author group, please list the individual authors and affiliations within this group in the acknowledgments section of your manuscript. Please also indicate clearly a lead author for this group along with a contact email address.

Additional Editor Comments (if provided):

Reviewers' comments:

Reviewer's Responses to Questions

**Comments to the Author**

1. Is the manuscript technically sound, and do the data support the conclusions?

Reviewer #1: Yes

Reviewer #2: Yes

2. Has the statistical analysis been performed appropriately and rigorously? 

Reviewer #1: Yes

Reviewer #2: Yes

3. Have the authors made all data underlying the findings in their manuscript fully available?

Reviewer #1: No

Reviewer #2: Yes

4. Is the manuscript presented in an intelligible fashion and written in standard English?

Reviewer #1: Yes

Reviewer #2: Yes

5. Review Comments to the Author

Reviewer #1: In this manuscript, Chloe Sarnowski and colleagues conducted a whole-genome sequence single-variant association and a gene-based rare/functional variant association studies to identify genomic regions associated with handgrip strength as a continuous variable. The authors discovered 10 loci from the single-variant analysis mostly in participants of AA and also identified one significant gene region from the gene-based analysis using EA sample. Although they used the high-quality sequence data, the sample size was fairly limited (N=13,552, even smaller in the stratified analyses) and the findings were not confirmed in an independent dataset (replication). The paper overall is well-written, and the authors well described and discussed the limitation of the study. Some minor issues could be addressed before publication.

Here are some things the authors could do to improve the clarity of the manuscript.

1. Abstract and Discussion: Please replace “15 new loci associated with handgrip strength” with “11 new loci …” since four gene regions from the gene-based test did not pass the significance level.

2. What's the rationale for adjusting for height in addition to BMI? Why different covariates were used in the UKBB sample (no sex interaction term)? Please explain in the Method section.

3. Were there any newly identified loci from the TOPMed AA analysis that had a significant association in UKBB EA participants (trans-ancestry loci)?

4. Please include the UKBB sample information in Table 1.

5. First paragraph of Result section: Did the authors actually test for handgrip strength differences by sex or race/ethnicities? Please provide a p value if the authors would like to use the term “lower”.

6. It would be helpful to include the phenotypic correlation between multiple handgrip traits in the manuscript (e.g. MEAN vs ONE).

7. Table 3: Suggest this goes in Supplementary.

Reviewer #2: 

“Leveraging handgrip strength measures and whole-genome sequences of six trans-ethnic studies, the authors associated 15 new loci with handgrip strength with rare and/or ethnicity-specific genetic variations. The statistical analyses were appropriately and rigorously performed using a set of advanced statistical methods. Their results support their conclusions. The manuscript has been well written. It is technically sound and reads interesting. For some potential improvements, I have a couple of comments/suggestions as below.

1. The participants used were from six study-reported ethnic groups, where African-Americans and Hispanics/Latinos have two and more ancestral populations. Some alleles are minor in one ancestral population but are major alleles in another ancestral population. It is unclear that how the authors defined “minor alleles”, i.e., in their principal component analyses (PCA) and relatedness estimates using the pooled genotypic data.

2. Participants from one ethnic group (i.e., European-Americans) would be unrelated to the participants from another ethnic group (i.e., African-Americans). Applying the pcrelate to pooled trans-ethnic study population would generate random relatedness estimates for numerous pairs of unrelated participants. It is unclear that how the authors avoid such random errors.   

3. For fine mapping in admixed populations, i.e., African-Americans and Hispanics/Latinos, adjusting for local ancestries proved a necessary and effective avenue to prevent false positives. Neglecting local ancestries could be one reason for the observed inflations in current results. It would be instructive to discuss or justify why not adjusting for local ancestries together with the global ancestries (PC1 – PC11). 

4. The authors adjusted for a small number of selected covariates in their analyses. Showing their significant effects with p-values would be informative to justify why they should be adjusted for. Besides, the authors could have neglected certain important latent confounders due to limited data availability. Applying some method/package (i.e., BACON) could help control the confounding of latent confounders.

5. For the single marker association results, presenting 95% concentration intervals (CIs) of the genomic inflation coefficients at 25%, 50% and 75% quantiles would be informative to show how well the confounders were controlled. Without such CIs, it is hard to judge how confident that the confounders were well controlled.”

6. PLOS authors have the option to publish the peer review history of their article (what does this mean?). If published, this will include your full peer review and any attached files.

Reviewer #1: No

Reviewer # 2: Yes

7a. If you answered "Yes" above, that you would like your identity to be public, please provide your full name, as it should appear on the published peer review. Please do not sign the review on behalf of another person.

Huaizhen Qin

---

## [Author Response · Author response to Decision Letter 0]

15 Apr 2021

Responses to the Reviewers' comments:

We would like to thank the reviewers for their thoughtful comments and appreciation of our work. We have responded below, in blue, to all reviewers’ comments and revised the manuscript accordingly.

Review Comments to the Author

Reviewer #1: In this manuscript, Chloe Sarnowski and colleagues conducted a whole-genome sequence single-variant association and a gene-based rare/functional variant association studies to identify genomic regions associated with handgrip strength as a continuous variable. The authors discovered 10 loci from the single-variant analysis mostly in participants of AA and also identified one significant gene region from the gene-based analysis using EA sample. Although they used the high-quality sequence data, the sample size was fairly limited (N=13,552, even smaller in the stratified analyses) and the findings were not confirmed in an independent dataset (replication). The paper overall is well-written, and the authors well described and discussed the limitation of the study. Some minor issues could be addressed before publication.

Here are some things the authors could do to improve the clarity of the manuscript.

1. Abstract and Discussion: Please replace “15 new loci associated with handgrip strength” with “11 new loci …” since four gene regions from the gene-based test did not pass the significance level.

We have revised these sentences in the manuscript accordingly.

2. What's the rationale for adjusting for height in addition to BMI? Why different covariates were used in the UKBB sample (no sex interaction term)? Please explain in the Method section.

We have included in our association analyses the same set of covariates as was used in the previous published GWAS of handgrip strength (Willems et al, Nat Commun, 2017). Height is a proxy for hand size and BMI is capturing the component of handgrip strength related to body size.

We thank the reviewer for noticing the discrepancy between the set of covariates included in our TOPMed and UK Biobank analyses. We have tested the significance of the two additional interaction terms in a null model (interaction of age and BMI with sex) in the UK Biobank. As both were strongly associated with handgrip strength, we have re-run the association analysis in the UK Biobank African-ancestry sample. We have revised the Supplementary Table with the replication results (now Supplementary Table 8) as well as the Method section describing the UK Biobank analyses.

3. Were there any newly identified loci from the TOPMed AA analysis that had a significant association in UKBB EA participants (trans-ancestry loci)?

The newly identified signals from the TOPMed AA analysis are specific to African-ancestry populations and do not exist or are very rare in European populations. Thus, they are not present in the Willems et al. EA handgrip strength GWAS summary statistics.

4. Please include the UKBB sample information in Table 1.

We thank the reviewer for this suggestion. We have modified Table 1 and Supplementary Tables 3 and 4 to include UKBB information and removed the descriptive Table of UKBB alone (previously Supplementary Table 13).

5. First paragraph of Result section: Did the authors actually test for handgrip strength differences by sex or race/ethnicities? Please provide a p value if the authors would like to use the term “lower”.

Following the reviewer’s suggestion, we performed formal t-tests to evaluate handgrip strength differences by sex and race/ethnicities.

Women versus Men:

t = 104.9, df = 7270.3, p-value < 2.2e-16

EA versus AA:

t = -11.266, df = 4868.2, p-value < 2.2e-16

However, as these differences could be confounded by different age and sex distributions in the EA and AA samples, we decided not to include these tests in the paper and to rephrase the first paragraph of the Result Section:

“Mean handgrip strength (SD) was equal to 23.5 kg (6.5) in women and 40.1 (9.9) in men. Mean age (SD) was 71 years (11) in EA participants and 60 years (18) in AA participants. Mean handgrip strength was equal to 28.9 kg (10.9) in EA participants and 31.6 kg (11.9) in AA participants.”

6. It would be helpful to include the phenotypic correlation between multiple handgrip traits in the manuscript (e.g. MEAN vs ONE).

We thank the reviewer for this suggestion. The overall handgrip strength phenotypic correlation between the MEAN and the ONE analysis was equal to 0.98. As several studies included in our analysis had only one exam where handgrip strength was available, we also calculated the phenotypic correlation by study. The handgrip strength phenotypic correlation between the MEAN and the ONE analysis was equal to 0.96, 0.95 and 0.93 for FHS, WHI and CHS respectively (the three studies for which multiple exams were available for handgrip strength). This information has now been added to the manuscript in the Methods section:

“The handgrip strength phenotypic correlation between the MEAN and the ONE analysis was equal to 0.96, 0.95 and 0.93 for FHS, WHI and CHS respectively”.

7. Table 3: Suggest this goes in Supplementary.

We have now moved Table 3 to the Supplement (now Supplementary Table 5). We have added one footnote to Table 2 to indicate which results were significant in the sex-stratified analyses and referred to Table 5 in S1 Text.

Reviewer #2: 

“Leveraging handgrip strength measures and whole-genome sequences of six trans-ethnic studies, the authors associated 15 new loci with handgrip strength with rare and/or ethnicity-specific genetic variations. The statistical analyses were appropriately and rigorously performed using a set of advanced statistical methods. Their results support their conclusions. The manuscript has been well written. It is technically sound and reads interesting. For some potential improvements, I have a couple of comments/suggestions as below.

1. The participants used were from six study-reported ethnic groups, where African-Americans and Hispanics/Latinos have two and more ancestral populations. Some alleles are minor in one ancestral population but are major alleles in another ancestral population. It is unclear that how the authors defined “minor alleles”, i.e., in their principal component analyses (PCA) and relatedness estimates using the pooled genotypic data.

We thank the reviewer for this comment. The variant calling in TOPMed has been performed jointly across all studies. The genotype files used for PCA were coded in terms of the same reference allele for all studies. We did not attempt to define minor alleles in any population, as this step is unnecessary for PCA and relatedness calculations. The effect allele frequency (EAF) presented in the manuscript has been calculated in the whole TOPMed sample.

2. Participants from one ethnic group (i.e., European-Americans) would be unrelated to the participants from another ethnic group (i.e., African-Americans). Applying the pcrelate to pooled trans-ethnic study population would generate random relatedness estimates for numerous pairs of unrelated participants. It is unclear that how the authors avoid such random errors.

We would like to clarify that participants from multiple ethnic groups are not necessarily unrelated. For example, the children of two people who identify with different ethnic groups may identify with either or both of the parents’ groups and will be related to people from each group. We have observed many cases of relationships between participants in different studies in TOPMed, so accurately accounting for relatedness between all participants in the analysis, and not just within study or ethnic group, is necessary to get appropriately calibrated association test statistics. PC-Relate was actually designed precisely for the scenario of estimating relatedness in a sample of individuals with multiple ancestries. PC-Relate adjusts for population structure (ancestry) among sample individuals through the use of ancestry representative principal components (PCs) to provide accurate relatedness estimates due only to recent family (pedigree) structure. It reports kinship values at or near zero for any pair of participants who are unrelated (in the sense of recent pedigree structure), and it will not produce random relatedness estimates for unrelated pairs of participants.

3. For fine mapping in admixed populations, i.e., African-Americans and Hispanics/Latinos, adjusting for local ancestries proved a necessary and effective avenue to prevent false positives. Neglecting local ancestries could be one reason for the observed inflations in current results. It would be instructive to discuss or justify why not adjusting for local ancestries together with the global ancestries (PC1 – PC11). 

We thank the reviewer for bringing up this point. We reviewed the literature on this topic (PMC3042179, PMC5079159, PMC5811405). These works indicate that local ancestry analysis is not required for variant discovery, but could be used to better inform follow-up analysis. Although we did not perform local ancestry analysis for this paper, it could be an avenue for future work to further investigate effects of the detected variants. We have added one sentence in the Discussion section:

“Finally, we have not considered in this paper local ancestry analysis that could help in the future to further investigate the effects of the detected variants.”

Also, we note that much of the literature on local ancestry assumes that SNP array data is used, so that LD patterns in the population studied have a strong effect on which variants are detected. Since our analysis is based on WGS, we think that local population stratification should not have biased the variant discovery.

4. The authors adjusted for a small number of selected covariates in their analyses. Showing their significant effects with p-values would be informative to justify why they should be adjusted for. Besides, the authors could have neglected certain important latent confounders due to limited data availability. Applying some method/package (i.e., BACON) could help control the confounding of latent confounders.

Following the reviewer’s suggestion, we have displayed in the word response document the significance of all covariates from the null model.

We also used the R package bacon to estimate bias and inflation corrected P-values for our main signals from the MEAN analysis, as suggested by the reviewer. All results passing the genome-wide threshold in our main analysis remained significant.

5. For the single marker association results, presenting 95% concentration intervals (CIs) of the genomic inflation coefficients at 25%, 50% and 75% quantiles would be informative to show how well the confounders were controlled. Without such CIs, it is hard to judge how confident that the confounders were well controlled.”

Following the reviewer’s suggestion, we calculated the genomic inflation factor lambda at the 25%, 50% and 75% quantiles for the MEAN analysis:

25% λ=1.078

50% λ=1.081

75% λ=1.092

It is not usual to calculate 95% confidence interval for the genomic inflation factor. We computed it for the median and the SE was very small: λ=1.081 95%CI: [1.080-1.082]. We expect the CI for the genomic inflation factor lambda at the 25% and 75% to be of similar range/magnitude.

---

## [Decision Letter · Decision Letter 1]

7 May 2021

PONE-D-20-39658R1

Identification of novel and rare variants associated with handgrip strength using whole genome sequence data from the NHLBI Trans-Omics in Precision Medicine (TOPMed) Program

PLOS ONE

Dear Dr. Sarnowski,

Thank you for submitting your manuscript to PLOS ONE. After careful consideration, we feel that it has merit but does not fully meet PLOS ONE’s publication criteria as it currently stands. Therefore, we invite you to submit a revised version of the manuscript that addresses the points raised during the review process.

We look forward to receiving your revised manuscript.

Kind regards,

Heming Wang, PhD

Academic Editor

PLOS ONE

Journal Requirements:

Reviewers' comments:

Reviewer's Responses to Questions

**Comments to the Author**

1. If the authors have adequately addressed your comments raised in a previous round of review and you feel that this manuscript is now acceptable for publication, you may indicate that here to bypass the “Comments to the Author” section, enter your conflict of interest statement in the “Confidential to Editor” section, and submit your "Accept" recommendation.

Reviewer #1: All comments have been addressed

Reviewer #2: (No Response)

2. Is the manuscript technically sound, and do the data support the conclusions?

Reviewer #1: Yes

Reviewer #2: Yes

3. Has the statistical analysis been performed appropriately and rigorously? 

Reviewer #1: Yes

Reviewer #2: Yes

4. Have the authors made all data underlying the findings in their manuscript fully available?

Reviewer #1: No

Reviewer #2: Yes

5. Is the manuscript presented in an intelligible fashion and written in standard English?

Reviewer #1: Yes

Reviewer #2: Yes

6. Review Comments to the Author

Reviewer #1: (No Response)

Reviewer #2: The authors inadequately addressed my comments as detailed below.

1. The participants used were from six study-reported ethnic groups, where African-Americans and Hispanics/Latinos have two and more ancestral populations. Some alleles are minor in one ancestral population but are major alleles in another ancestral population. It is unclear that how the authors defined “minor alleles”, i.e., in their principal component analyses (PCA) and relatedness estimates using the pooled genotypic data.

New Comment 1

Replying this comment, the authors said they did not did not attempt to define minor alleles in any population. In their revision, however, the authors still used a “minor allele frequency (MAF)” threshold of 0.01 in their description of PCA and used “minor allele count” in Table 3. They should clarify these in some way, i.e., either replacing “minor allele” with a better phrase, or clarify in what a sense they defined minor allele.

2. Participants from one ethnic group (i.e., European-Americans) would be unrelated to the participants from another ethnic group (i.e., African-Americans). Applying the pcrelate to pooled trans-ethnic study population would generate random relatedness estimates for numerous pairs of unrelated participants. It is unclear that how the authors avoid such random errors.

New comment 2

In their response to this comment, the authors stated that the pcrelate reported kinship values “near” zero for pairs of unrelated participants. Such near-zero estimates are very common and what I meant by “random relatedness”. The package they used might have used some arbitrary truncation threshold to determine pairs of unrelated persons. Explicitly stating the parameters used can be informative for the readers.

3. For fine mapping in admixed populations, i.e., African-Americans and Hispanics/Latinos, adjusting for local ancestries proved a necessary and effective avenue to prevent false positives. Neglecting local ancestries could be one reason for the observed inflations in current results. It would be instructive to discuss or justify why not adjusting for local ancestries together with the global ancestries (PC1 – PC11).

New comment 3:

To reply this comment, the authors reviewed several papers on local-ancestry adjustments (PMC3042179, PMC5079159, PMC5811405). Such papers actually proved the necessity of local-ancestry adjustments for fine mapping in admixed populations. In particular, for WGS data, it is instructive to consider local ancestries in addition to global ancestries for rare variant association mapping, see Sci Rep. 2019 Apr 1;9(1):5458; PMC6443736, Nature Genetics volume 44, pages243–246(2012), and Nature Genetics volume 53, pages195–204(2021).

Moreover, the averages of accurate genome-wide local ancestry proportions represent global ancestries much more accurately than do the major pcair PCs. By the authors’ Table 3, only PC1 and PC10 appeared significantly relevant. Even if I did not see the joint distributions of the major pcair PCs, I am frankly not sure if the pcair PCs well represent ancestries, see also my New Comment 4. When applying for the GENESIS package, tuning parameters must be carefully selected to determine the unrelated sets and the relatives set. I did not see the values and the justification.

4. The authors adjusted for a small number of selected covariates in their analyses. Showing their significant effects with p-values would be informative to justify why they should be adjusted for. Besides, the authors could have neglected certain important latent confounders due to limited data availability. Applying some method/package (i.e., BACON) could help control the confounding of latent confounders.

New comment 4

The authors provided Table 4 to show the handgrip-covariates associations under 3 distinct models. Multiple covariates, especially PC2, …, PC9, and PC11, appeared to be insignificant under each of the 3 models. Assuming the major PCs well represent ancestries – sufficiently capturing both between- and within-subpopulation structure. Jointly adjusting for study IDs and PCs might be unnecessary or even problematic, as indicated by the insignificance of multiple study IDs and PCs. Some PC plots stratified by study IDs might be informative to demonstrate the connections between PCs and study IDs.

Latent confounders would be common and it is great to see BACON corrected P-values for the 10 SNPs passing genome-wide significance threshold. But I did not see any details of the adopted BACON parameters or the Q-Q plots of the BACON corrected p-values. The default parameters of the BACON package are often inappropriate and should be tuned carefully. See my New Comment 5.

5. For the single marker association results, presenting 95% concentration intervals (CIs) of the genomic inflation coefficients at 25%, 50% and 75% quantiles would be informative to show how well the confounders were controlled. Without such CIs, it is hard to judge how confident that the confounders were well controlled.

New Comment 5

The authors appeared to misunderstand the above comment and derived λ=1.081 as the inflation factor for the median with 95% confidence interval [1.080-1.082]. Per this result, I am 95% confident the true inflation factor is larger than 1 since the lower confidence limit is larger than 1, aka, showing significant inflation.

A concentration interval is different from a confidence interval. They have different definitions and computation formulae. The number of effective tests plays critical roles on the computation of the concentration interval for a genomic inflation factor. Let their analysis be valid and let the proportion of the true association signals be much smaller than 0.25. Given the very large number of effective tests, the genomic inflation factors at the 25%, 50% and 75% quantiles should be much closer to 1 than the authors showed. In other words, the concentration intervals are very narrower with lower limits < 1 and upper limits > 1.

7. PLOS authors have the option to publish the peer review history of their article (what does this mean?). If published, this will include your full peer review and any attached files.

Reviewer #1: No

Reviewer #2: **Yes: **Huaizhen Qin

---

## [Author Response · Author response to Decision Letter 1]

26 May 2021

We have provided clarifications to all the reviewer’s questions or concerns in a separate document and highlighted in blue.

---

## [Decision Letter · Decision Letter 2]

9 Jun 2021

Identification of novel and rare variants associated with handgrip strength using whole genome sequence data from the NHLBI Trans-Omics in Precision Medicine (TOPMed) Program

PONE-D-20-39658R2

Dear Dr. Sarnowski,

We’re pleased to inform you that your manuscript has been judged scientifically suitable for publication and will be formally accepted for publication once it meets all outstanding technical requirements.

Kind regards,

Heming Wang, PhD

Academic Editor

PLOS ONE

Additional Editor Comments (optional):

Reviewers' comments:

Reviewer's Responses to Questions

**Comments to the Author**

1. If the authors have adequately addressed your comments raised in a previous round of review and you feel that this manuscript is now acceptable for publication, you may indicate that here to bypass the “Comments to the Author” section, enter your conflict of interest statement in the “Confidential to Editor” section, and submit your "Accept" recommendation.

Reviewer #2: (No Response)

2. Is the manuscript technically sound, and do the data support the conclusions?

Reviewer #2: Yes

3. Has the statistical analysis been performed appropriately and rigorously? 

Reviewer #2: Yes

4. Have the authors made all data underlying the findings in their manuscript fully available?

Reviewer #2: Yes

5. Is the manuscript presented in an intelligible fashion and written in standard English?

Reviewer #2: Yes

6. Review Comments to the Author

Reviewer #2: The authors have satisfactorily addressed most of my comments. Here are several minor comments for them to carefully reword relevant discussions.

The study IDs are coarse proxies of between-study structure and can sometimes be highly associated with PCs (Figure 1 from the authors). Apart from ancestral components and cryptic relatedness, there are other confounders in genetic association studies, such as genotyping platforms and other technical drivers. Per my inspection, the PC-AiR algorithm does explicitly separate such confounders.

It is subjective to use the rule of thumb on genomic inflation factors <= 1.1 as reported in a book chapter. The concept of 95% concentration band was not only used in methodological studies [1,2] but also used in real-world large-scale GWASs [3-5] to show rigorous validity.

References

1. Cao, S., Qin, H., Gossmann, A., Deng, H.-W. & Wang, Y.-P. Unified tests for fine-scale mapping and identifying sparse high-dimensional sequence associations. Bioinformatics 32, 330-337 (2016).

2. Huaizhen, Q., Zhao, J. & Zhu, X. Identifying Rare Variant Associations in Admixed Populations. Scientific Reports (Nature Publisher Group) 9(2019).

3. Cho, Y.S., et al. A large-scale genome-wide association study of Asian populations uncovers genetic factors influencing eight quantitative traits. Nature genetics 41, 527-534 (2009).

4. Satake, W., et al. Genome-wide association study identifies common variants at four loci as genetic risk factors for Parkinson's disease. Nature genetics 41, 1303 (2009).

5. Weedon, M.N., et al. Genome-wide association analysis identifies 20 loci that influence adult height. Nature genetics 40, 575 (2008).

7. PLOS authors have the option to publish the peer review history of their article (what does this mean?). If published, this will include your full peer review and any attached files.

Reviewer #2: **Yes: **Huaizhen Qin

---

## [Editor Report · Acceptance letter]

24 Jun 2021

PONE-D-20-39658R2 

Identification of novel and rare variants associated with handgrip strength using whole genome sequence data from the NHLBI Trans-Omics in Precision Medicine (TOPMed) Program 

Dear Dr. Sarnowski:

I'm pleased to inform you that your manuscript has been deemed suitable for publication in PLOS ONE. Congratulations! Your manuscript is now with our production department. 

Kind regards, 

on behalf of

Dr. Heming Wang 

Academic Editor

PLOS ONE